# Dataset Transformation System for Sign Language Recognition Based on Image Classification Network

Sang-Geun Choi , Yeonji Park and Chae-Bong Sohn *

Department of Electronics and Communications Engineering, Kwangwoon University, Seoul 01897, Korea
* Correspondence: cbsohn@kw.ac.kr

**Abstract:** Among the various fields where deep learning is used, there are challenges to be solved in motion recognition. One is that it is difficult to manage because of the vast amount of data. Another is that it takes a long time to learn due to the complex network and the large amount of data. To solve the problems, we propose a dataset transformation system. Sign language recognition was implemented to evaluate the performance of this system. The system consists of three steps: pose estimation, normalization, and spatial–temporal map (STmap) generation. STmap is a method of simultaneously expressing temporal data and spatial data in one image. In addition, the accuracy of the model was improved, and the error sensitivity was lowered through the data augmentation process. Through the proposed method, it was possible to reduce the dataset from 94.39 GB to 954 MB. It corresponds to approximately 1% of the original. When the dataset created through the proposed method is trained on the image classification model, the sign language recognition accuracy is 84.5%.

**Keywords:** motion recognition; dataset transformation system; sign language recognition; spatial–temporal map (STmap); image classification model

## 1. Introduction

Sign language is a method for the deaf to communicate in society by using a variety of information, such as hand movements and facial expressions. In general, the ultimate goal of sign language recognition is to communicate with others by interpreting a deaf individual's words or sentences. Even if the sign language performs the same action, it may be a different word or sentence depending on the range of movement [1]. In recent years, there has been an increasing interest in deep learning applied to various fields, and it has contributed to technological improvement [2]. There have recently been numerous studies in the field of sign language recognition using deep learning to classify images or videos. The reason we chose sign language recognition is because it has both the characteristics of motion recognition and the characteristics of a time series language translation. Deep learning models that classify images have low complexity compared to models that classify videos [3]. The reason is that the model used for video classification uses a time series learning model based on recurrent neural networks (RNN) [4]. The RNN-based classification model is used for the classification of data with continuity, such as natural language processing (NLP) and motion recognition. Natural language processing requires general sentence data, and the video data is required for learning for motion recognition.

Deep learning models require large amounts of data for high accuracy [5]. In the case of natural language processing, even if the amount of data increases, the physical capacity occupied by the data does not increase rapidly. However, in the case of motion recognition, the video data are used for learning. When the amount of video data increases, the capacity increases rapidly. There are challenges to be solved in the field of motion recognition using a relatively high complexity model and a large capacity dataset. First, large amounts of data and high complexity increase the learning time. Second, it costs a lot to manage the data.

In contrast to the existing sign language recognition system, which recognizes using a time series classification model, we propose a dataset transformation system to enable

sign language recognition with a low complexity image classification model. The focal point of the proposed method is the process of converting video data into images. The amount of data can be effectively reduced only when information about human motion is extracted from the video data. Through pose estimation, it is possible to extract human skeleton information from video data [6]. In order to extract the skeleton information in a consistent range, normalization was performed. In the video that contains human movement, the person does not always move at the center of the video or at a specific location. This process is necessary because we cannot limit the position of a person. Normalized data are converted to data between 0 and 1 so that the position of the person is not affected. Subsequently, the STmap (spatial–temporal map) is generated by converting these data to the RGB coordinate system. Through this process, the system can create a dataset in a new type that can be trained on the image classification model.

However, the lack of a dataset remains a problem in the sign language recognition problem. So, we perform data augmentation in order to train on a small amount of data. Data augmentation is a technique widely used when training deep learning models, and it is possible to increase the amount of data to increase the learning accuracy [7]. This paper uses two data augmentation techniques: frame skipping and salt-and-pepper noise.

In this paper, we propose a data transformation system that converts video data into images and augments the data to train on an image classification model. Because the image classification model is used, the learning complexity is reduced, which is expected to reduce the learning time and increase the accuracy. In addition, since the video data are converted into an image, the data capacity is also expected to decrease.

The main contributions of this work are as follows:

- Proposal of a process that enables fast and accurate motion recognition by converting video data into image data.
- Proposal of a process that enables efficient data management by reducing capacity through data transformation.
- Proposal of a training method for a model that is not sensitive to pose estimation errors by adding noise.

This paper is organized as follows. Section 2 is about the motion recognition, sign language recognition and spatial–temporal map as related work. Section 3 describes the configuration of the dataset transformation system. Section 4 describes the results of the experiment, and the last two sections are discussion and conclusion.

## 2. Related Work

### 2.1. Motion Recognition

Motion recognition is to determine what kind of action a person performs based on information acquired through a camera or other sensors. There are two types of motion recognition: a method of attaching a sensor or other devices to the body, and an image-based method captured by a camera. A lot of research on image-based motion recognition is in progress. Sensor-based motion recognition classifies human motion by analyzing the body coordinate vector obtained from the sensor [8,9]. It is cumbersome to attach a sensor to the body for motion capture, but accurate information can be obtained. Video-based motion recognition analyzes videos captured by a camera and classifies human motions [10–12]. There are various approaches to video-based motion recognition. Like the method of extracting coordinates through a motion capture suit, a method of extracting a person's key point coordinates from a video through a pose estimation process and analyzing these data for the motion recognition. There is a method of recognizing the motion by analyzing the characteristics of the image. Keypoints extracted by pose estimation have lower accuracy than the coordinates obtained through the motion capture sensor, but it is possible to eliminate the hassle of wearing it. A lot of research is being performed on the method of using 3D information by combining images and sensors [13]. The 2D image is acquired by the camera, and the depth information is obtained through the sensor and used as 3D information.

### 2.1.1. CNN-Based Motion Recognition

CNN is a model that classifies images by extracting features of a single image. Several approaches are being attempted to use CNN for motion recognition. The first approach extracts the features of each frame through CNN and then fuses time information [14]. Ref. [14] tried to learn features using a 2D CNN model and input multiple frames using various fusion methods for temporal information. Using the 2D CNN structure is advantageous for learning because it is possible to use excellent models pretrained with the ImageNet dataset.

Figure 1 shows the fusion of temporal data in a 2D CNN in several approaches. The bottom of Figure 1 shows the video sequence data of the input layer, and the red, green, and blue boxes represent the convolution, normalization, and pooling layers, respectively. The top two layers are the fully connected layers. The data fusion method is divided into late fusion, early fusion, and slow fusion depending on the time of fusion. According to [14], the late fusion performed better than the other methods. The second approach extends the convolution operation to the time domain [15]. To overcome the limitations of 2D CNNs and learn temporal features, many studies have been conducted on 3D CNN structures [16]. By adding time to a 2D image, it is used as 3D data, and convolution is performed with a 3D filter to learn temporal features. However, by using a 3D filter, the number of parameters to learn is much larger than that of a 2D CNN.

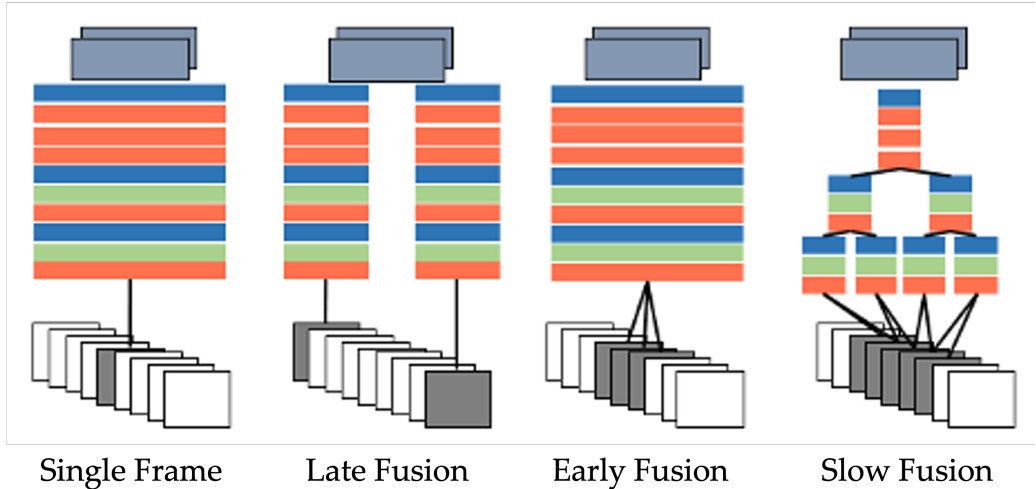

| Single Frame | Late Fusion | Early Fusion | Slow Fusion |

**Figure 1.** Temporal data fusion method to CNN.

The third approach is to encode the video into a dynamic image containing spatial and temporal information and then apply a CNN for image-based recognition [17]. Ref. [17] uses the rank pooling concept to encode video frames into dynamic images with spatial–temporal features. The generated image is trained on a CNN-based network to perform motion recognition. The fourth approach decouples the two elements and adopts a multi-stream network [18]. This approach trains the temporal and spatial parts, respectively, and passes the SoftMax layer at the end. A spatial stream receives each frame of a video as an input like a general image classification model. The temporal stream receives an optical flow as an input. Optical flow expresses the change of motion between frames.

### 2.1.2. RNN-Based Motion Recognition

RNN is a model designed to analyze sequential data [19]. Sequential data refer to any data that contain elements which are ordered into sequences. The basic RNN structure has a disadvantage in that when the length of the sequence becomes long, past information cannot be transmitted backward [20]. The model designed to solve this problem is the LSTM (long short-term memory) model. RNN-based models have been widely used for natural language processing. Many studies have been conducted to use the RNN-based

model for motion recognition. One of those studies is to use a model that combines CNN and RNN. Figure 2 shows the structure of the model in which CNN and RNN are combined.

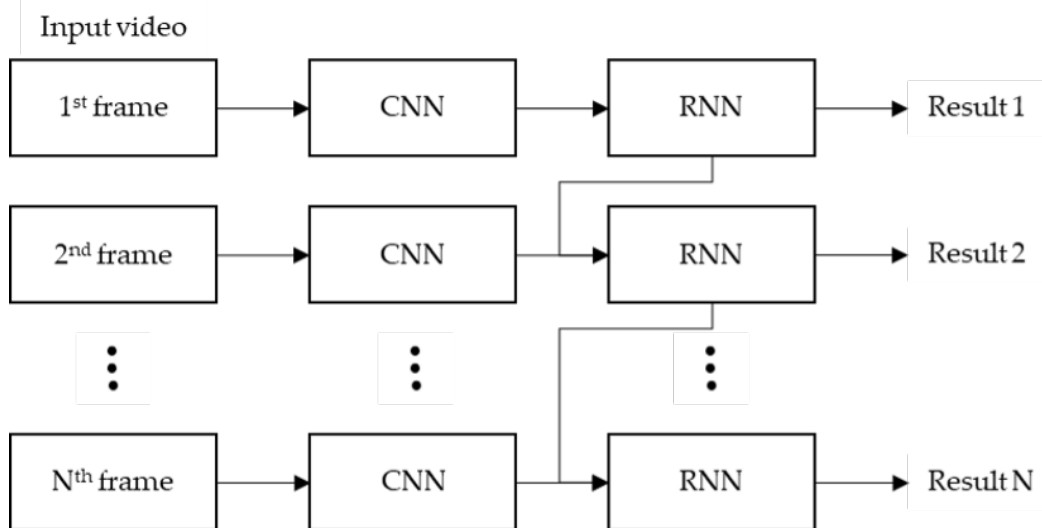

**Figure 2.** Structure of a motion recognition model using CNN-RNN model.

First, the spatial information of each frame of a video is trained by CNN. By using the results of the trained CNN as an input to the RNN, motion recognition is enabled. Since then, various approaches have been attempted, such as combining a RNN model with a CNN-based motion recognition model [21].

*2.2. Sign Language Recognition*

Sign language is a visual language for the deaf or hearing impaired to communicate using hand gestures, facial expressions, and other information. Language translation using artificial intelligence has been used in various places. To translate sentences by learning the relationship between words, most translations use RNN-based deep learning networks [22]. In the case of spoken language, words or sentences are used for learning. As a model for translating sign language, a combination of RNN for natural language processing and CNN for image feature analysis is often used.

Sign language recognition was studied using a model combined with CNN and RNN structures in [23–26]. There is also a study on sign language recognition by combining the RNN-based GRU model with the pose estimation process [27]. In [27], human skeleton information was extracted by pose estimation, and this information was used as input to the GRU. In addition, in order to increase the accuracy of translation, research is being conducted to increase the weight of words frequently used in specific situations by applying context awareness [28,29].

*2.3. Pose Estimation*

Pose estimation is a technique for detecting the position and orientation of an object in computer vision. The human body is made up of several joints. Through pose estimation, we can extract the positions of human joints and various feature points [6]. Pose estimation methods include a top-down method and a bottom-up method. The top-down method detects a person in an image and then crops the image. Pose estimation is performed from the cropped image [30,31]. The bottom-up method extracts the keypoints of a person in the image first. Pose estimation is performed by analyzing the relationship between the extracted keypoints [30,31].

Both methods have their advantages and disadvantages. The top-down method has higher accuracy than the bottom-up method, but when there are several people, it takes a long time because pose estimation is performed from all detected people. Although the bottom-up method has relatively low accuracy, it can be processed in real time because

there is no process of detecting a person. The bottom-up method was selected for real-time translation, and the OpenPose library, which is a bottom-up approach, was applied to the experiment.

OpenPose

OpenPose is a real-time pose estimation library that jointly detects keypoints of the human body, hands, face, and feet in a single image [32,33]. Figure 3 is a figure expressing keypoints that can be extracted through OpenPose.

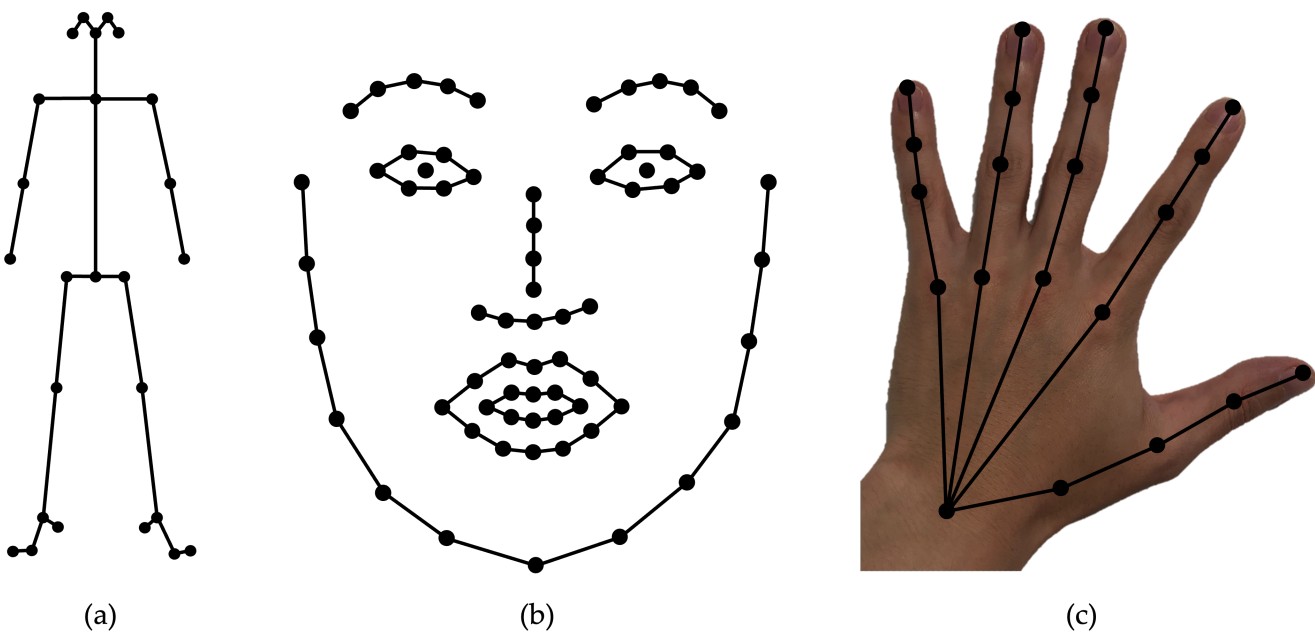

(a)      (b)      (c)

**Figure 3.** Keypoints extracted using OpenPose : (**a**) 25 keypoints extracted from the body; (**b**) 70 keypoints extracted from face; (**c**) 21 keypoints extracted from one hand.

OpenPose extracts 135 keypoints from the human body. The keypoint on the wrist is used as the same keypoint on the hand and body. So, 135 keypoints are extracted [32,33]. Each keypoint has three pieces of information. These are the $x$-axis coordinates, the $y$-axis coordinates, and the accuracy of the coordinate extraction. The $x$-axis and $y$-axis coordinate values range from 0 to the horizontal and vertical lengths of the video, and the coordinate extraction accuracy has a value between 0 and 1. The sign language recognition model is trained by focusing on the keypoints of the person. The background of the video is data that are not necessary for training.

## 3. Materials and Methods

The method proposed in this paper used a new type of image created through several processes for learning. This chapter consists of a description of the overall structure of the proposed method and a description of each element constituting the system.

### 3.1. Dataset Transformation System Structure

The proposed method in this paper was designed with the structure shown in Figure 4. Then, the system was applied to the image classification model and used for sign language recognition.

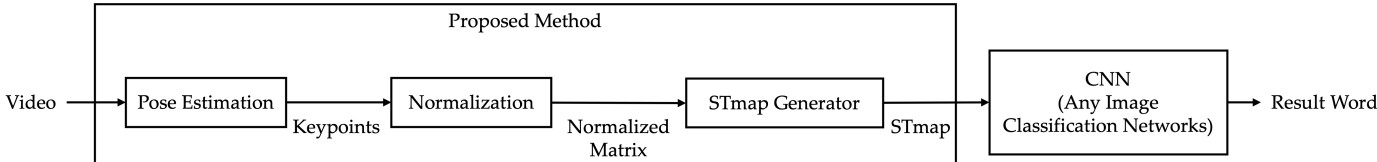

**Figure 4.** The structure of the proposed method and its application to the sign language recognition model.

First, when sign language video is input, keypoints are extracted through the pose estimation process. In order to use a certain level of keypoints in all data, the normalization proceeded as the next step. Normalized data were converted into the proposed image in this paper through image generation. The generated image dataset is used as an input to the image classification model to learn. This allows us to enable motion recognition in a general image classification model.

### 3.2. Normalization

In the video of the dataset used for training, the person performing the sign language is always centered. However, when the learned model is applied to the real environment, it is impossible to fully control the position of the person performing the sign language. Assume that person A and person B perform the same sign language action. If person A is on the left side of the video and person B is on the right side of the video, the Y coordinates will be the same, but the X coordinate values will be different. Normalization was performed to solve these problems. In addition, the data standardization process was performed for comparison. After the normalization and standardization process, the values of the coordinates change to a value between 0 and 1. A comparison of the images generated through the two methods will be explained in the STmap generator chapter.

### 3.3. Keypoint Pixel Mapping

When the normalization process is completed, the extracted keypoints are processed into matrix data. To convert these data into an image, the coordinate values and the values indicating the accuracy of the coordinates were mapped to RGB values, respectively. Since the coordinates and the accuracy of coordinates included in the matrix have a value between 0 and 1, each value is multiplied by 255 to convert it to a value suitable for the color coordinate system. The normalized or standardized coordinates of the extracted keypoints are matched by converting the X coordinate to the R value of RGB in Equation (1) and the Y coordinate to the G value of RGB in Equation (2).

$$R = X' or X'' \times 255, \tag{1}$$

$$G = Y' or Y'' \times 255, \tag{2}$$

$$B = C \times 255, \tag{3}$$

where $X'$ and $Y'$ represent normalized coordinates, and $X''$ and $Y''$ are standardized coordinates. $C$ represents the extraction accuracy of the coordinates. Equation (3) is a formula for converting the $C$ value to the $B$ value of RGB.

### 3.4. STmap Generator

A spatial–temporal map (STmap) was created to make the data converted from the x–y coordinate system to the color coordinate system into an image with a certain rule. Figure 5 represents the information input to each pixel constituting the image.

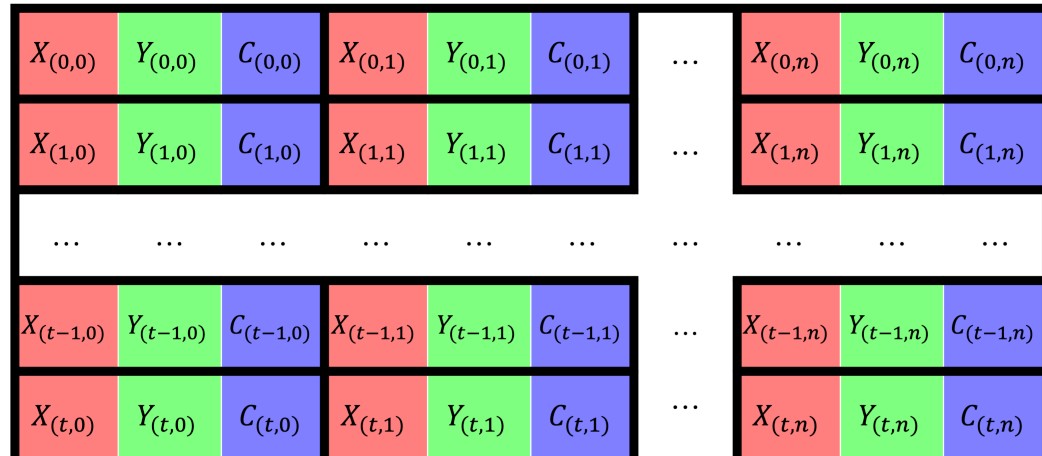

**Figure 5.** Construction of STmap images generated by the proposed method.

One pixel in the STmap image corresponds to one extracted keypoint. The horizontal axis constituting the image is a series of keypoints, and one row consists of 135 keypoints extracted from the body. The order is from left to right, that is, left hand (21 keypoints), right hand (21 keypoints), face (70 keypoints) and body (25 keypoints). The vertical axis means time, and moving to the next row is the same as moving to the next frame of the video. Figure 6 shows the result of converting the word 'burn' in Korean sign language into an STmap image.

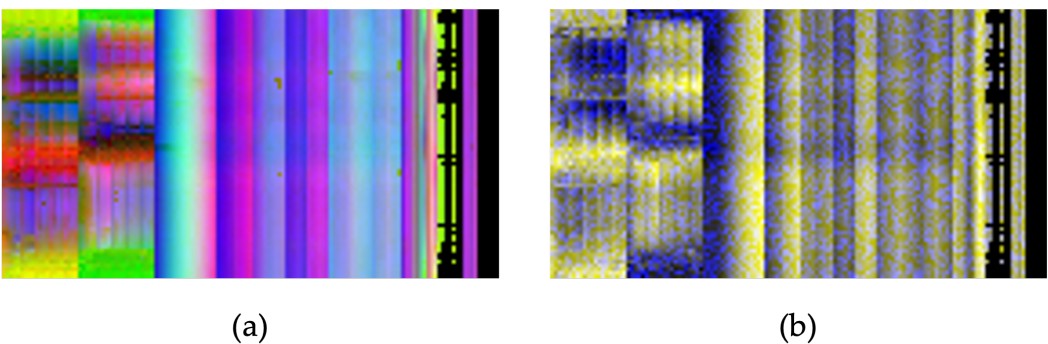

(a)          (b)

**Figure 6.** Images generated by the proposed method (word: burn); (**a**) the result of the normalization process; (**b**) the result of the standardization process.

In Figure 6, the color of the image changes as the movement of a person appears. The change becomes more noticeable in the image generated as the result of normalization. As shown above, the characteristics of the generated image are vivid. The image pattern changes a lot due to the large movement of both hands because of the nature of the sign language. Facial movement contains important information in sign language, but the pattern did not appear strongly in the image, so the keypoints of the face were excluded in this experiment.

### 3.5. Data Augmentation

One of the important factors in training a deep learning model is the size of the dataset. In the case of motion recognition, it takes a lot of time compared to other datasets because a person directly performs a motion, and it should be recorded as a video. In order to replenish a small dataset, the data augmentation process is used for training many deep learning models. In this experiment, we performed data augmentation in two ways: frame skipping and salt and pepper, focusing on the general method for high accuracy among state-of-the-art video augmentation techniques [34].

### 3.5.1. Frame Skipping

Video consists of multiple frames of images. Although every frame contains important information of the video, very few frame drops are not a significant problem for viewing. We used frame skipping for data augmentation rather than reducing training or prediction latency. Figure 7 shows how the original video is split into two videos through frame skipping.

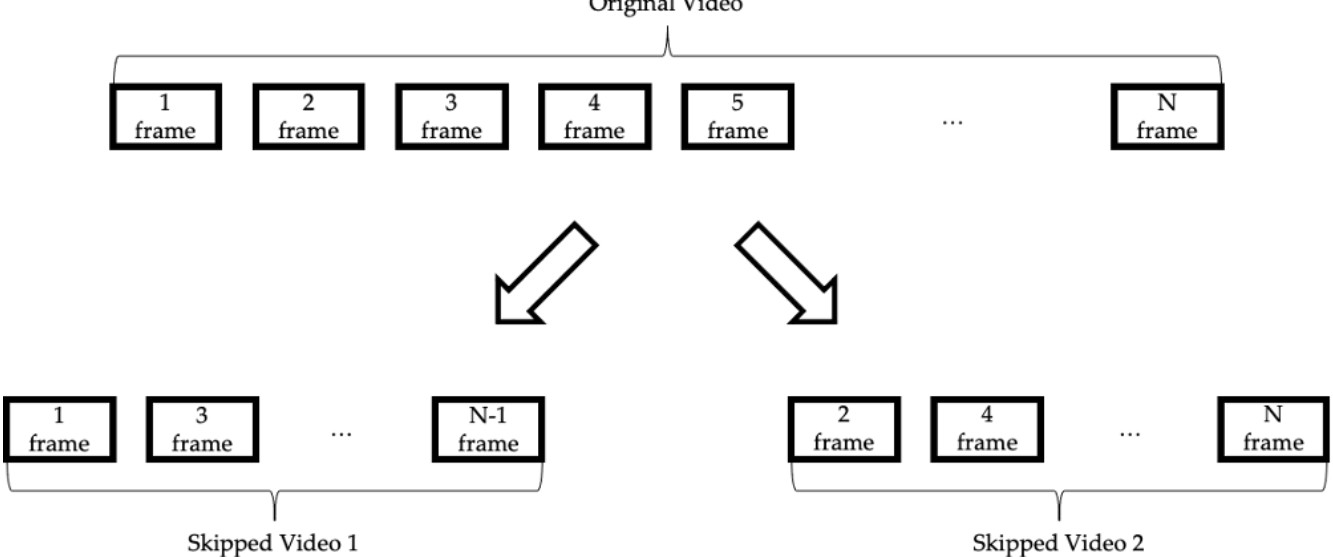

**Figure 7.** Frame skipping process at 2 frame intervals.

Assuming that the original video is 60 fps, the two videos separated by 2 frames are 30 fps. When the frame rate drops, smooth viewing is difficult, but there is no problem in obtaining information about the video. However, in terms of data, the two videos divided by frame skipping are completely different data. Figure 8 shows the results of applying the frame-skipping process to the training data at five frame intervals.

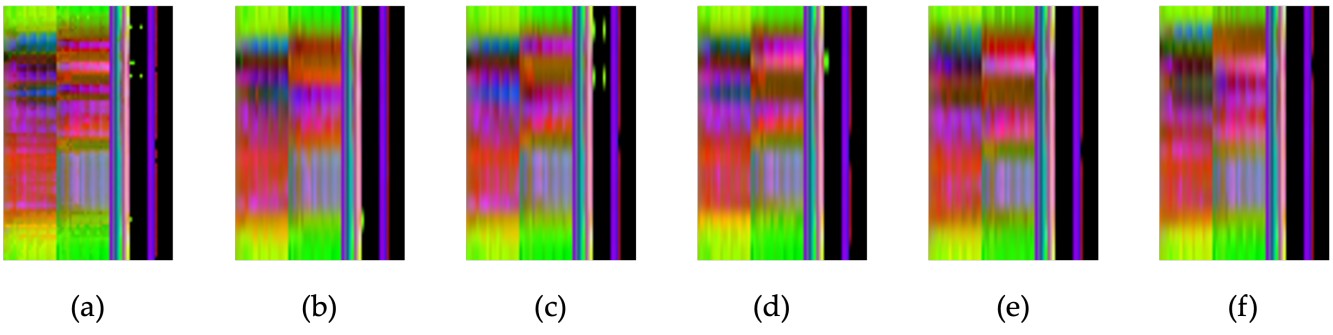

| (a) | (b) | (c) | (d) | (e) | (f) |

**Figure 8.** Result of frame skipping at 5 frame intervals; (**a**) original data; (**b**–**f**) frame skipped data.

When the original data and the frame-skipped data were compared, the patterns appearing in the original image were confirmed in all images.

### 3.5.2. Salt and Pepper Noise

The method proposed in this paper uses a human keypoint by converting it into a color coordinate system. Salt-and-pepper noise causes random pixel values in an image to be (0, 0, 0) or (255, 255, 255). If this noise is applied to the proposed method, it creates a situation where the coordinates of the keypoint are extracted incorrectly during the pose-estimation process. In the case of a model that uses images for training, noise is

intentionally added to increase the amount of the dataset and it makes this model more resistant to noise. Figure 9 shows the result of applying salt-and-pepper noise.

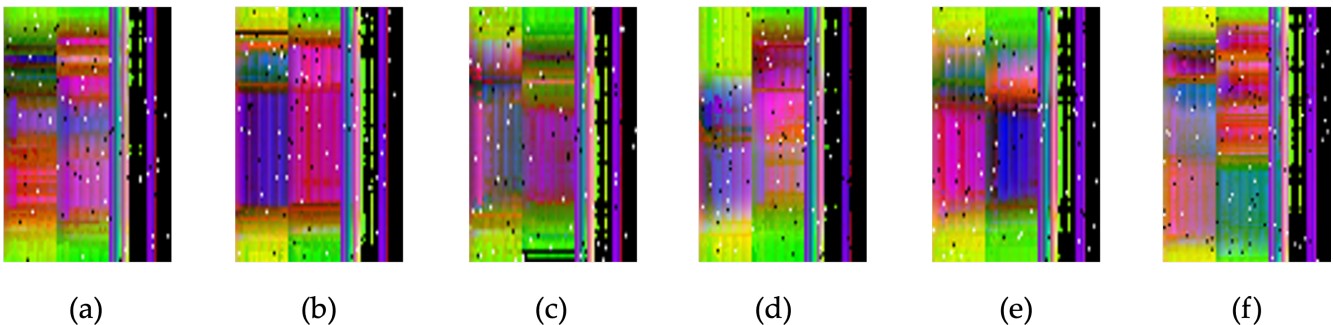

(a)  (b)  (c)  (d)  (e)  (f)

**Figure 9.** Image with salt-and-pepper noise applied to the word; (**a**) highway; (**b**) heart attack; (**c**) fever; (**d**) bleeding; (**e**) bomb; (**f**) fire.

### 3.6. Sign Language Dataset

The dataset used in this paper is the Korean sign language dataset provided by KETI (Korea Electronics Technology Institute). The words constituting the dataset consist of words and sentences that can be used in an emergency situation. Table 1 shows the number of words and sentences included in the KETI sign language dataset. Ten sign language experts performed actions on a total of 514 words and sentences, and a dataset was created by capturing the front and side from various angles with two cameras. So, to demonstrate one word and sentence, it is comprised of 20 videos.

**Table 1.** Number of words and sentences in the KETI sign language dataset.

|  | Word | Sentence |
| --- | --- | --- |
| Number of words (sentences) | 419 | 105 |
| Number of videos (front angle) | 4190 | 1050 |
| Number of videos (side angle) | 4190 | 1050 |
| Number of videos | 8380 | 2100 |

Figures 10 and 11 lists some video captures from the KETI sign language dataset.

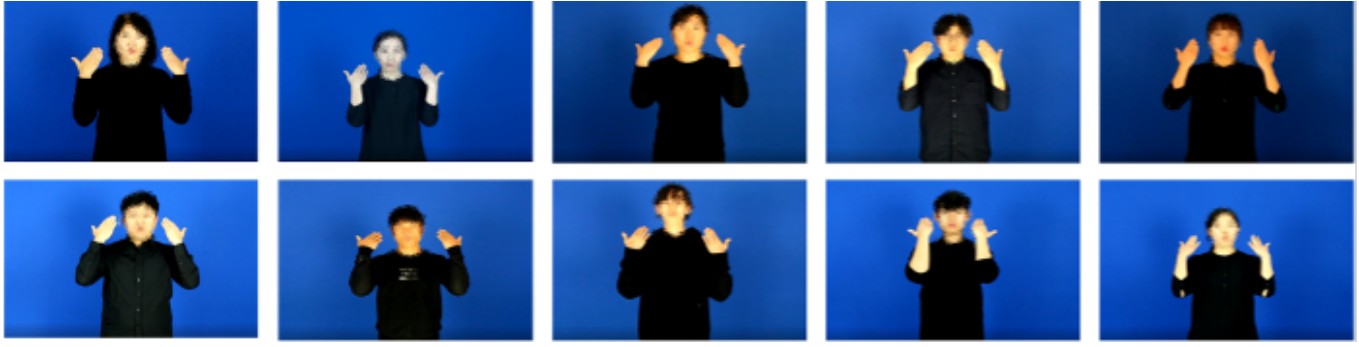

**Figure 10.** Examples of sign language datasets representing "school" word performed by 10 signers in the front angle.

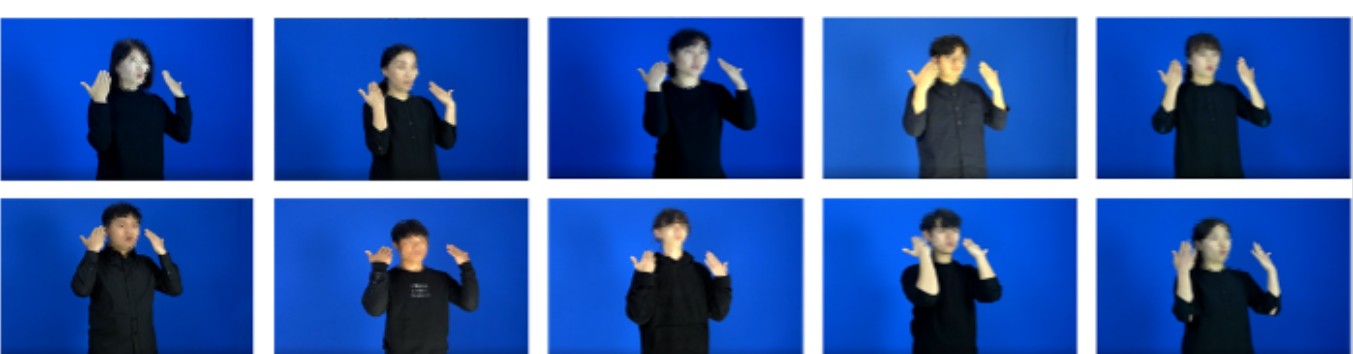

**Figure 11.** Examples of sign language datasets representing "school" word performed by 10 signers in the side angle.

## 4. Results

We conducted an experiment to confirm the two results. The first measured how much dataset size was saved when the proposed method was applied. To train a CNN-based image classification model, we converted the video data into the image data. In this process, a decrease in data capacity occurred, and the decrease in capacity was a great advantage in data management. The second measured the recognition accuracy when this dataset was applied to the image classification model. Even if management becomes easier with a smaller capacity, it is not valuable if the data are not suitable for training the model. In the process of measuring the learning accuracy, two types of experiments were conducted. First, it was an experiment for comparison before and after data augmentation using only four-word data. Second, learning was performed on 80% of the training set among the entire dataset, and the accuracy was measured for 20% of the test set. ResNet training took a total 120 epochs, and the learning rate was 0.00002.

### 4.1. Experiment Environments and Dataset Transformation Results

In the following system environment, we performed model training and verification experiments. We trained the ResNet model in cuda 10.1 using the Python programming language. The detailed experimental environment is given in Table 2. As shown in Table 1, our dataset in this paper consists of a total of 10,840 videos with 1920 × 1080 pixels composed of words and sentences expressed by 10 signers, which is an open-source sign language dataset obtained from KETI.

**Table 2.** System environments.

| System Environment | |
|---|---|
| CPU | AMD Ryzen 5 5600X 6-core Processor |
| RAM | 48 GB |
| VGA | NVIDIA GeForce RTX 3060 |
| OS | Windows 10 |
| Tools | CUDA 10.1 |
| CNN Model | ResNet |

Table 3 compares the capacity occupied by dataset. We confirmed that our proposed dataset transformation method was 11.1 KB before data augmentation and 120 KB after data augmentation when we performed it on single data with an average original size of 2.7 MB. When looking at only single data, it was possible to reduce the capacity by 4% compared to the original.

**Table 3.** Number of words and sentences in the KETI sign language dataset.

|  | Single Data | Entire Dataset |
| --- | --- | --- |
| Original | 2.7 MB | 94.39 GB |
| Proposed method without data augmentation | 11.1 KB | 105 MB |
| Proposed method with data augmentation | 120 KB | 954 MB |
| Ratio | 4% | 1% |

When we applied our proposed method to the entire dataset, with a total capacity of 94.39 GB, it was confirmed that it totaled 105 MB before and 120 MB after data augmentation. In the entire dataset, it was possible to reduce the capacity by 1% compared to the original.

*4.2. Sign Language Recognition Results*

The generated dataset was trained on ResNet, an image classification model, and its accuracy was measured. To check the effect of data augmentation, the accuracy before and after application was compared. In the experiment comparing the effect of data augmentation, only four words were used for learning. Figure 12 shows the validation loss during training.

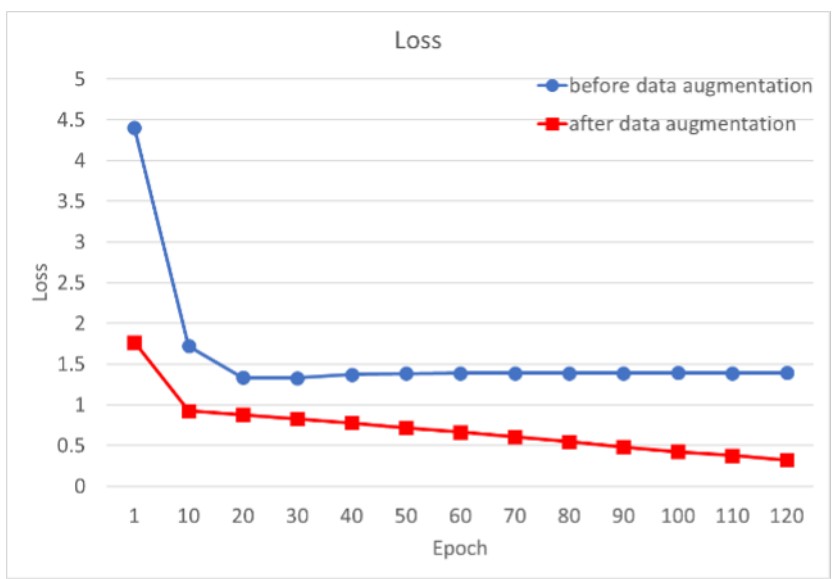

**Figure 12.** Validation Loss.

It was confirmed that the loss decreased in both graphs before and after the application of data augmentation. As depicted in Figure 12, the graph before data augmentation began with an initial loss of about 4.5 and converged to a value of about 1.5 after about epoch 20, making accurate classification difficult. However, after data augmentation, the graph revealed that the initial loss began around 1.75 and continued to decrease until the last epoch. This demonstrates that the data augmentation technique we employed has a positive impact on training. Figure 13 is the confusion matrix for the test data after training.

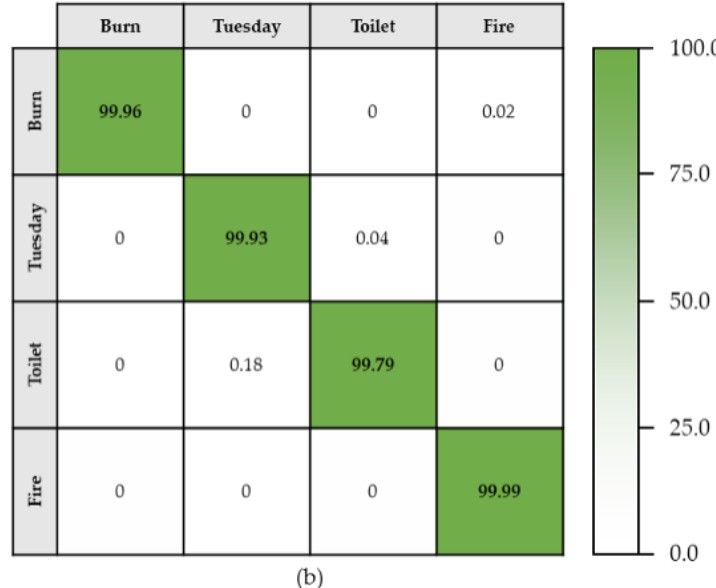

**Figure 13.** Confusion matrix of four words before and after data augmentation; (**a**) before data augmentation; (**b**) after data augmentation.

The difference between the results before and after data augmentation is clear. When data augmentation is not applied, the amount of data used for training is 16 per word. On the other hand, the number after data augmentation is 192 per word. The results after data augmentation show about 99% accuracy in all data. Table 4 compares the accuracy measured by using the entire sign language data for training in the proposed method with the accuracy of the other two models.

**Table 4.** Comparison of training accuracy of the proposed method, CNN-RNN and GRU models.

| Model | Accuracy (%) |
|---|---|
| Proposed method with ResNet | 84.5 |
| CNN-RNN [35] | 24.0 |
| GRU with OpenPose [27] | 52.8 |

When the image data generated through the proposed method were trained on ResNet, the accuracy was 84.5%. A model using CNN and RNN combined and a model using OpenPose and GRU were used for the comparison model. Both models are used for motion recognition, and the second model is designed for sign language recognition. As a result of training the sign language video dataset on the CNN-RNN model, the accuracy was measured to be 24.0%. The model used for sign language recognition by extracting human keypoints from the GRU model had an accuracy of 52.8%.

## 5. Discussion

In the research for motion recognition, a large amount of data and an RNN-based model with relatively high complexity were used. There also exists a motion recognition model using CNN, but it has a higher complexity than a model for classifying 2D images. In this paper, we proposed a dataset transformation system to improve this situation. Various attempts were made to solve these problems. Among them, there were approaches that added the pose estimation process, and it was used as an input for the RNN-based model. Although the size of the input was reduced by adding pose estimation process, it still required high complexity by using an RNN-based model. We conducted the experiment

by adding the posture estimation process as used in previous studies. In the previous study, the results of posture estimation were directly used as input to the RNN-based model, but we added the process of transforming it into a spatial–temporal map. By adding the transformation process, it was possible to use a CNN-based model rather than an RNN-based model with relatively high complexity. This enabled fast and accurate learning. In addition, by adding a transformation process, it was possible to reduce the capacity of the data used for learning. The proposed method was able to reduce the amount of dataset to about 1% and confirmed the relatively high accuracy compared to other studies using the same dataset. However, this study still has several problems to be solved. The first is that the size of the input layer of the image classification model is fixed, but the length of the video is different. In sign language, there are not only words expressed with one action, but also words that should be expressed with a combination of several actions. Simple motions relatively shorten the length of the video, and complex motions lengthen it. This problem is more noticeable in the case of sign language sentences. The second is that one datum is associated with one word or sentence. That is, in the case of a sentence including the trained word, it may be recognized as a word rather than a sentence. If these problems are resolved, this study is expected to develop into a model that can translate sign language beyond sign language recognition.

## 6. Conclusions

In summary, we proposed a process of mapping a video dataset to STmap and transforming it into an image dataset. The results were confirmed by applying the proposed method to sign language recognition. This study was conducted to solve two problems occurring in the field of motion recognition. The first problem was that it was difficult to manage huge volumes of data. In the case of video data used for motion recognition, it is not necessary to require the information of all areas. We started investigating how to extract only the necessary information needed and realized that motion recognition is proceeded with a focus on people. In order to extract only human information, the pose estimation process was applied. The dataset created by the proposed method occupies only 1% of the capacity of the original dataset. We extracted the whole keypoints, including face, body, and hands, but in the case of sign language recognition, only keypoints of body were used to perform learning. It is expected that data management will be easier if data are selected in consideration of the purpose of a certain motion classification. The second problem is to use a time series model with relatively high complexity. We made it possible to convert data from video into images in the process of converting data. This image has information about keypoints as well as information about time. Since data of all time are included in one image, it is possible to sufficiently determine the features with an image classification model with relatively low complexity. It was confirmed through experiments that sign language recognition was performed with higher accuracy, particularly when comparing to the time series model used for sign language recognition. In this experiment, training was performed using ResNet, but if an advanced image classification model is used, higher accuracy can be expected in further research.

**Author Contributions:** Conceptualization, S.-G.C. and Y.P.; methodology, S.-G.C.; software, Y.P.; validation, S.-G.C., C.-B.S. and Y.P.; formal analysis, C.-B.S.; investigation, Y.P.; resources, C.-B.S.; data curation, Y.P.; writing—original draft preparation, S.-G.C.; writing—review and editing, S.-G.C.; visualization, S.-G.C.; supervision, C.-B.S.; project administration, S.-G.C.; funding acquisition, C.-B.S. All authors have read and agreed to the published version of the manuscript.

**Funding:** This work was supported by the Ministry of Education of the Republic of Korea and the National Research Foundation of Korea (NRF-2020S1A5B8101323).

**Institutional Review Board Statement:** Not applicable.

**Informed Consent Statement:** Not applicable.

**Data Availability Statement:** Not applicable.

**Conflicts of Interest:** The authors declare no conflict of interest.

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
