# Peer review of "Dataset Transformation System for Sign Language Recognition Based on Image Classification Network"

_applsci, doi:10.3390/app121910075_

Round 1
Reviewer 1 Report (Previous Reviewer 2)
First of all, I would like to note that the article deals with a challenging problem with a significant impact on the life of deaf people and their families. So, from this perspective, the conducted research is highly essential.
Focusing on the submission, apart from the red highlighted sentences, it is unclear if other additions/corrections have been made compared to the previous version of the manuscript.
However, some observations-suggestions of major concern for the paper's improvement are the following:
The introduction section should be enriched with some paragraphs assuming more references. For example, the authors could elaborate on the motivation and the impact of the current research.
The novel aspects of the proposed method should be highlighted in the Introduction or at the end of the Related Work.
The dataset should be presented more clearly, giving more details about the signers, specific words and sentences used. A statistical description of the dataset words is necessary.
The authors should explain which words-sentences are captured in SL in Figure 10.
Figure 11, the validation loss, should be analysed in relation to the topic under consideration and the involved parameters. The authors should investigate which parameters most affect it and to what extent.
Figure 12 needs to be discussed more using terms from Machine Learning. Moreover, the confusion matrix can give useful information about the words (class labels) that the model confuses recognizing them as similar, leading to classification errors. In other words, it gives information/explains the obtained recognition errors. These are some aspects that should be incorporated into the manuscript.
The Accuracy metric should be defined for the current classification problem. Apart from accuracy, based on the confusion matrix some other common metrics from ML should be captured in a table, such as recall, precision and properly analyzed concerning the topic under consideration. Is the problem treated as a multi-class classification task? Please define and clarify it.
Also, why the authors focused on ResNet (advantages)? what does this model serve in the problem attempted to solve? Which other models could be used in each place?
Sections 4.1 and 4.2 are too short. The authors could properly merge these sections into a single one and extend their content by discussing more in Tables 2 and 3.
The discussion section should be extended by comparing the current study with previous works (similarities, differences).
The conclusions section should be extended giving some future extensions of the current study.
Minor corrections:
The font size is different in sections 5 and 6. Please correct it accordingly.
Equations (1), (2), and (3) aren't referred to inside the text.
Author Response
Thank you for your review.
Most of what you pointed out has been corrected.
We would like to respond to some of the points you pointed out that do not apply.
Review : The Accuracy metric should be defined for the current classification problem. Apart from accuracy, based on the confusion matrix some other common metrics from ML should be captured in a table, such as recall, precision and properly analyzed concerning the topic under consideration. Is the problem treated as a multi-class classification task? Please define and clarify it.
Answer : It is important to use new data sets to determine accuracy. However, we are dealing with the process of creating a new type of dataset itself as an important topic. For this reason, we only want to show information about accuracy.
Review : Also, why the authors focused on ResNet (advantages)? what does this model serve in the problem attempted to solve? Which other models could be used in each place?
Answer : For the same reason as the previous answer. The new type of dataset can be applied to any image classification model. Applying a modern image classification model will of course give good accuracy, but I don't think it's important at this stage. We hope that a new type of dataset will be released and many studies will proceed.
Thanks again for your review.

Reviewer 2 Report (New Reviewer)
This paper considers the problem of motion recognition deep learning models. The authors develop a data transformation method with the three steps of pose estimation, normalization and spatial temporal map generation. In general, the authors consider an important problem in the data pre-processing in AI models. Please see the following comments.
1. In the Introduction part, please discuss the problems with the exiting works on the sign language recognition models. What is the problem with the existing models? For which type of images the exiting models are not accurate enough?
2. From Fig. 4, the reviewer’s understanding is that the proposed method is a feature extraction module for the CNN. So, please highlight why the existing feature extraction networks/methods (like ResNet, Alexnet, etc.) still need to be improved?
3. In Section 3.6 (Frame skipping), why you consider frame skipping? To reduce prediction latency or training latency. Obviously, the training latency can be ignored because this can be done offline. For prediction latency, is your application target at real-time video analytics? Please note reducing the frame number normally reduces the prediction accuracy.
4. In the same subsection, the motivation to use pepper and salt is not justified. Why you consider those pepper and salts may happen in practical cases? How about other augmentation methods like blurness, color variation, etc.
5. In Section 4, you collected the data-set yourself or crawled from open-source data-set?
6. In Figure 12, why the confusion matrix only has 4 classes? The sign language should have many types.
Author Response
Thank you for your review.
Most of what you pointed out has been corrected.
We would like to respond to some of the points you pointed out that do not apply.
Review : 6. In Figure 12, why the confusion matrix only has 4 classes? The sign language should have many types.
Answer : We tested whether a new type of dataset could be applied to an image classification model using four words. We think that it is important that the learning accuracy of the model is high, but we wrote the paper thinking that it is more important whether a new type of data set has research value. We would appreciate it if you could focus on the new type of dataset rather than the accuracy of the image classification model and proceed with the review.
Thanks again for your review.

Round 2
Reviewer 1 Report (Previous Reviewer 2)
The authors tackled most of the suggestions. They have made an attempt to improve the content of the manuscript but the article needs some minor (still important) enhancements.
1) The English language writing style must be improved. For example, lines 389-391 in Discussion ("Various attempts were made to solve these problems. In various papers, many attempts have been made to") should be rewritten. Check the whole text for similar issues.
2) The information in the discussion section should be improved to be more understandable. Also, Discussion should highlight similarities and differences between the current study and previous works.
3) Table 2 is not mentioned in the text. Concerning this table, alternatively, the authors could incorporate its information in a paragraph.
Author Response
Thank you for your review.
I checked the entire paper again and corrected the English expression as much as possible. I'd appreciate it if you could point it out if I'm missing something.
In the discussion section, corrections and additions have been made to clarify differences and similarities with previous studies.
I also added a sentence referring to Table 2.
Thank you for your help that has improved the quality of the paper.
Thanks again for your review.

Reviewer 2 Report (New Reviewer)
Thanks to the authors for addressing the comments and including the revisions. The reviewer has no further major comments. Please proofread the manuscript and improve the presentation if this manuscript can be accepted for publication. Besides, please format the paper using Latex and also output the figures in pdf/eps format to present a nicely-written manuscript.
Author Response
Thank you for your review.
The paper was rewritten in Latex. Also, convert the result to pdf and attach it.
Thank you for your help that has improved the quality of the paper.
Thanks again for your review.
Thanks again for your review.

This manuscript is a resubmission of an earlier submission. The following is a list of the peer review reports and author responses from that submission.
Round 1
Reviewer 1 Report
The authors use a dataset related to sign language translation but use it as a SL recognition.
The related works section is very poor. Both subsections are not enough defined. Subsection 2.1 must describe existing methods to summarise video data different to STMaps or the methods that inspire STMaps approach. Subsection 2.2 does not include detailed information about previous methods related to Sign Language Translation or Recognition. Figure 1 is not interesting at this point.
Section 3 is a mixture of Related works, specific tools and methodology. It must be reorganized. Figures 2, 3 and 4 and 5 are very basic. Normalization and standardization formulas are known and do not need to be shown.
The more interesting and valuable contribution is the STMap definition but has some flaws, could you show how a short sign appears? I assume it would be practically black. That means that the STMap depend on the duration of the sentence, and even if the sentence is very long more than one image would be needed, how to deal with that problem?
The approach is interesting but the paper does not have a scientific structure and much more work is needed. Limitations must be shown and code is desirable to be shared to reproduce the results.
Author Response
Dear reviewer
Thank you for your detailed review.
I will write it in the form of a response to the content you pointed out.
Point 1 : The authors use a dataset related to sign language translation but use it as a SL recognition.
Answer 1 : Recognizing that the proposed method is closer to sign language recognition than sign language translation, we modified the expression of sign language translation to sign language recognition.
Point 2 : The related works section is very poor. Both subsections are not enough defined. Subsection 2.1 must describe existing methods to summarise video data different to STMaps or the methods that inspire STMaps approach. Subsection 2.2 does not include detailed information about previous methods related to Sign Language Translation or Recognition. Figure 1 is not interesting at this point.
Answer 2 : Added content for related work. The content of STmap was written with the content that both temporal and spatial information are used for learning during motion recognition part. Added content for Figure 1 (modified as Figure 2).
Point 3 : Section 3 is a mixture of Related works, specific tools and methodology. It must be reorganized. Figures 2, 3 and 4 and 5 are very basic. Normalization and standardization formulas are known and do not need to be shown.
Answer 3 : Moved the Pose Estimation part from Section 3 to Section 2. Figures 2 and 3 have been removed. Figures 4 and 5 are basic but necessary and have been maintained. Removed normalization and standardization formulas.
Point 4 : The more interesting and valuable contribution is the STMap definition but has some flaws, could you show how a short sign appears? I assume it would be practically black. That means that the STMap depend on the duration of the sentence, and even if the sentence is very long more than one image would be needed, how to deal with that problem?
Answer 4 : The existing discussion has been changed to a conclusion. Section 5 discussion has been added and the problem to be addressed by this study has been drawn up.
Point 5 : The approach is interesting but the paper does not have a scientific structure and much more work is needed. Limitations must be shown and code is desirable to be shared to reproduce the results.
Answer 5 : We wanted to solve the problem with a very simple approach. It is true that there is a lack of scientific structure in the process. We will solve the problems you pointed out through continuous research in the future.
Thanks again for your review.

Reviewer 2 Report
The authors study a challenging problem that targets the deaf community. It is proposed a dataset transformation system on which SL translation is performed. The system consists of three steps: posture estimation, normalization, and STmap generation. The model shows improved accuracy and the error sensitivity was lowered through the data augmentation process.
The authors should improve and clarify many aspects.
The Introduction section is too short and doesn't sufficiently define the general framework of the study.
The contribution of the paper isn't highlighted in a convincing manner at the end of the Introduction. The introduction section needs enrichment. The article is missing relevant references to the problem.
The review of the state-of-the-art isn't sufficient. It lacks recent and relevant works in the area of SL recognition and translation. Some recent works that could take into consideration
- Context-Aware Automatic Sign Language Video Transcription in Psychiatric Interviews (Sensors 22): https://doi.org/10.3390/s22072656
- Continuous Sign Language Recognition through a Context-Aware Generative Adversarial Network (Sensors 21): https://doi.org/10.3390/s21072437
- A Hierarchical Ontology for Dialogue Acts in Psychiatric Interviews: https://doi.org/10.1145/3453892.3461349
The presented methodology and the mathematical DL models aren't captured properly. Please elaborate on them and explain how these models facilitate the purpose of the study.
Why did the authors select these specific words to classify through the proposed system? Do they relate to a specific context that concerns deaf people? The authors should justify their choice.
The authors have incorporated too many images. Please refine them and reduce their size as the text content is disproportionate to the number of images.
The Discussion section should be renamed as Conclusions.
A Discussion section, before conclusions, is necessary and should highlight the key findings of your study by pointing to the novelty of this paper, and the similarities and differences of other ongoing studies in this field. Also, it should discuss any limitations of the work during the research design and evaluation.
Also, the technical terms haven't been explained in detail. The authors omitted necessary information about the environment, frameworks and experiment setup.
Finally, there is no intuitive commentary on the results.
Author Response
Dear reviewer
Thank you for your detailed review.
I will write it in the form of a response to the content you pointed out.
The authors study a challenging problem that targets the deaf community. It is proposed a dataset transformation system on which SL translation is performed. The system consists of three steps: posture estimation, normalization, and STmap generation. The model shows improved accuracy and the error sensitivity was lowered through the data augmentation process.
The authors should improve and clarify many aspects.
Point 1 : The Introduction section is too short and doesn't sufficiently define the general framework of the study.
Answer 1 : The contents of the intro have been modified and the framework of study has been added.
Point 2 : The contribution of the paper isn't highlighted in a convincing manner at the end of the Introduction. The introduction section needs enrichment. The article is missing relevant references to the problem.
Answer 2 : The contents of the contribution of this paper have been corrected. Added a reference to the problem that the complexity of RNN-based models used for motion recognition is relatively high.
Point 3 : The review of the state-of-the-art isn't sufficient. It lacks recent and relevant works in the area of SL recognition and translation. Some recent works that could take into consideration
1. Context-Aware Automatic Sign Language Video Transcription in Psychiatric Interviews (Sensors 22): https://doi.org/10.3390/s22072656
2. Continuous Sign Language Recognition through a Context-Aware Generative Adversarial Network (Sensors 21): https://doi.org/10.3390/s21072437
3. A Hierarchical Ontology for Dialogue Acts in Psychiatric Interviews: https://doi.org/10.1145/3453892.3461349
The presented methodology and the mathematical DL models aren't captured properly. Please elaborate on them and explain how these models facilitate the purpose of the study.
Answer 3 : In addition to the three papers above, the research objectives and methodologies of several other papers in the field of sign language recognition and translation have been added to Section 2 Related Work.
Point 4 : Why did the authors select these specific words to classify through the proposed system? Do they relate to a specific context that concerns deaf people? The authors should justify their choice.
Answer 4 : Added reasons for choosing sign language to Section 1 Introduction.
Point 5 : The authors have incorporated too many images. Please refine them and reduce their size as the text content is disproportionate to the number of images.
Answer 5 : The number of images was maintained. However, unnecessary images were removed and an image of the result was added.
Point 6 : The Discussion section should be renamed as Conclusions.
Answer 6 : The existing Section 5 discussion was changed to Section 6 conclusion.
Point 7 : A Discussion section, before conclusions, is necessary and should highlight the key findings of your study by pointing to the novelty of this paper, and the similarities and differences of other ongoing studies in this field. Also, it should discuss any limitations of the work during the research design and evaluation.
Answer 7 : A new section 5 discussion has been created.
Point 8 : Also, the technical terms haven't been explained in detail. The authors omitted necessary information about the environment, frameworks and experiment setup.
Answer 8 : Insufficient explanation was added, and it was not previously written because it was not judged that the experimental environment was important, but it was added because it was thought necessary.
Point 9: Finally, there is no intuitive commentary on the results.
Answer 9 : An intuitive commentary of the results was added.
Thanks again for your review.

Reviewer 3 Report
The paper addresses the challenging and interesting problem of sign langage recognition, by proposing a « dataset transformation system ».
Unfortunately, there are several problems with the paper.
The authors mention several times « sign langage translation », for example in introduction : « To measure the accuracy of the proposed method, it was applied to sign language translation. » Here we are more in the context of sign langage recognition, translation is far more complicated and assumes that we can extract sentences from sign langage utterrances.
The writing has to be improved.
Several sentences are awkward « This paper intends to propose a method to solve the problems that appear in the field of motion recognition. », as if the paper would solve the problem of motion recognition. Please, be more precise and make it clear what are the problems solved by the paper.
Some sentences lack precision. For example in introduction :
- « As a result, it was possible to sufficiently reduce the amount of dataset through the proposed system. What is « sufficiently » ?
- « In addition, the learning accuracy is high, so it is not inferior to practical use. « Pllease, be more precise.
In the paper, when it is mentioned « dataset transformation system » it would be preferrable to use « data representation » since this is a tranform of RGB images towards a more compact representation of the signer in order to simplify the subsequent machine learning.
The usefulness and quality of some of the figures is questionnable. In addition, the reference of figure 4 is not mentioned, and I know for sure that it is not the authors’figures.
The originality of the paper has to be better explained. For example, in terms of representation, it should be better underlined what are the contributions of the paper compared to existing spatio-temporal representations that use 2D spatio-temporal concatenation of openpose joints, such as Huy-Hieu Pham, Houssam Salmane, Louahdi Khoudour, Alain Crouzil, Pablo Zegers, Sergio A Velastin. “Spatio-Temporal Image Representation of 3D Skeletal Movements for View-Invariant Action Recognition with Deep Convolutional Neural Networks”. Special Issue on Deep Learning-Based Image Sensors, Intelligent Sensors, Vol. 19 (8), 2019
The results are limited, they do not show :
- in which extent the data augmentation (with noise) improves the results. For that, the ResNet should be tested with and without data augmentation.
- whether the performances (using Resnet) are better or worse with or without the proposed representation
Author Response
Dear reviewer
Thank you for your detailed review.
I will write it in the form of a response to the content you pointed out.
The paper addresses the challenging and interesting problem of sign langage recognition, by proposing a « dataset transformation system ».
Unfortunately, there are several problems with the paper.
Point 1 : The authors mention several times « sign langage translation », for example in introduction : « To measure the accuracy of the proposed method, it was applied to sign language translation. » Here we are more in the context of sign langage recognition, translation is far more complicated and assumes that we can extract sentences from sign langage utterrances.
The writing has to be improved.
Answer 1 : Recognizing that the proposed method is closer to sign language recognition than sign language translation, we modified the expression of sign language translation to sign language recognition.
Point 2 : Several sentences are awkward « This paper intends to propose a method to solve the problems that appear in the field of motion recognition. », as if the paper would solve the problem of motion recognition. Please, be more precise and make it clear what are the problems solved by the paper.
Answer 2 : The problem to be solved has been modified so that it can be seen more clearly.
Point 3 : Some sentences lack precision. For example in introduction :
- « As a result, it was possible to sufficiently reduce the amount of dataset through the proposed system. What is « sufficiently » ?
- « In addition, the learning accuracy is high, so it is not inferior to practical use. « Pllease, be more precise.
Answer 3 : Ambiguous sentences were corrected and removed.
Point 4 : In the paper, when it is mentioned « dataset transformation system » it would be preferrable to use « data representation » since this is a tranform of RGB images towards a more compact representation of the signer in order to simplify the subsequent machine learning.
Answer 4 : It contains the process of maintaining and managing datasets by transforming video data into images to preserve expression.
Point 5 : The usefulness and quality of some of the figures is questionnable. In addition, the reference of figure 4 is not mentioned, and I know for sure that it is not the authors’figures.
Answer 5 : Omitting external image references is a big problem, thanks for pointing it out. We added an image reference.
Point 6 : The originality of the paper has to be better explained. For example, in terms of representation, it should be better underlined what are the contributions of the paper compared to existing spatio-temporal representations that use 2D spatio-temporal concatenation of openpose joints, such as Huy-Hieu Pham, Houssam Salmane, Louahdi Khoudour, Alain Crouzil, Pablo Zegers, Sergio A Velastin. “Spatio-Temporal Image Representation of 3D Skeletal Movements for View-Invariant Action Recognition with Deep Convolutional Neural Networks”. Special Issue on Deep Learning-Based Image Sensors, Intelligent Sensors, Vol. 19 (8), 2019
Answer 6 : The existing discussion has been changed to a conclusion. Section 5 discussion was newly written to express the originality of the paper.
Point 7 : The results are limited, they do not show :
- in which extent the data augmentation (with noise) improves the results. For that, the ResNet should be tested with and without data augmentation.
- whether the performances (using Resnet) are better or worse with or without the proposed representation
Answer 7 : Comparison results before and after data augmentation were prepared. During the comparison, the result of better learning was measured and the image was corrected. A comparison before and after application of the proposed method was prepared through Table 4. It is impossible to compare with the same model because the data types before and after using the proposed method are different.
Thanks again for your review.

Round 2
Reviewer 2 Report
The authors have made a significant effort to improve the quality of the manuscript.
I would suggest the authors proofread the manuscript and check the English language style and writing.
Also, the authors should check if the use of proposition 'of' in the title retains the meaning they want to give.
A recommended title is the following: Dataset Transformation System for Sign Language Recognition Based on Image Classification Network
Author Response
Thank you for your positive review.
As you suggested, "of" in the title of the paper has been changed to "based on".
In addition, English expressions and grammar were corrected.
There are corrections and additions made by other reviewers' comments.
Thanks again for your review.

Reviewer 3 Report
Dear authors,
The paper has been modified after the first round of reviews. Unfortunately, there are still many changes to do before resubmitting.
In addition, some of the reviewers' comments have not been taken into account.
Generally speaking, the writing and figures could be improved. To my mind, some figures are not eough explained and some of them are not useful. Several sentences lack precision.
In the experiments, it is really surprising that the methods [35] and [27] get so poor performance.
Additional experiments should be conducted : STMap with RNN-CNN, Stmap + GRU, since the SOTA methods that are chosen are based on these models.
Other suggestions :
Abstract :
- "And sign" --> remove And
- Please introduce the signification of STmap (Spatio-temporal map)
- " When the dataset created through the proposed method is trained on the image classification model, the sign language recognition accuracy is 84.5%." --> what was the initial accuracy, without augmentation?
- " As a result, it was possible to reduce the amount of dataset through the proposed method." --> These sentence is not useful or not precise enough
In the article:
- "Sign language is the ability to communicate through hand movements." : No. Sign language use also body, space, face.
- "The key point extracted"--> keypoints
- Fig 1 : Please precise what the colors refer to. If I understand well, the input is at the bottom and the output is at the top. It is not straightforward because arrows are missing.
Fig 1 is not well explained in the text.
- Fig 2 is one example of the use of CNN + RNN. Is it your figure, your proposal? If not please put a reference of a paper that uses this model.
- "Sign language communicates using hand gestures, not sounds." -> No (see above)
- This language is used for the deaf. --> this should be said earlier
- "A language is a combination of words and words to form a sentence. "
- Figure 3, please put the references of figure 3 (b) and (c)
- with the structure show in Figure 4.--> shown
- p6 : Figure 4 should be simplified by putting all the steps on one unique line.
- process proceeds --> redundant
- Figure 5 is not useful. The normalization can be understood without image.
- Section 4 should be better written. The beginning is abrupt
Author Response
Thank you for your review.
I will write it in the form of a response to the content you pointed out.
Dear authors,
The paper has been modified after the first round of reviews. Unfortunately, there are still many changes to do before resubmitting.
In addition, some of the reviewers' comments have not been taken into account.
Generally speaking, the writing and figures could be improved. To my mind, some figures are not eough explained and some of them are not useful. Several sentences lack precision.
Point 1 : In the experiments, it is really surprising that the methods [35] and [27] get so poor performance.
Answer 1 : In the case of sign language recognition through the structure of [35], it is the result obtained by our own experiment. We tried to get the highest possible accuracy by modifying hyperparameters, but 24% was the best. In the case of [27], the accuracy of the test set described in the paper was taken and written.
Point 2 : Additional experiments should be conducted : STMap with RNN-CNN, Stmap + GRU, since the SOTA methods that are chosen are based on these models.
Answer 2 : As far as I am aware, RNN-CNN models and GRU models are used for learning sequence data. However, STmap has time information, but not sequence data. What I want to do with STmap is to enable motion recognition using an image classification model.
Other suggestions :
Abstract :
Point 3 : - "And sign" --> remove And
Answer 3 : Deleted the word "And".
Point 4 : - Please introduce the signification of STmap (Spatio-temporal map)
Answer 4 : We wrote that STmap is an abbreviation of Spatial-Temporal Map.
Point 5 : - " When the dataset created through the proposed method is trained on the image classification model, the sign language recognition accuracy is 84.5%." --> what was the initial accuracy, without augmentation?
Answer 5 : Added comparison of results before and after data augmentation. When comparing before and after data augmentation, the results are obtained from some data, not all data.
Point 6 : - " As a result, it was possible to reduce the amount of dataset through the proposed method." --> These sentence is not useful or not precise enough
Answer 6 : The sentence you pointed out has been removed.
In the article:
Point 7 : - "Sign language is the ability to communicate through hand movements." : No. Sign language use also body, space, face.
Answer 7 : I wrote that various information such as facial expressions as well as hand movements are required.
Point 8 : - "The key point extracted"--> keypoints
Answer 8 : Corrected "The key point extracted" to "Keypoints".
Point 9 : - Fig 1 : Please precise what the colors refer to. If I understand well, the input is at the bottom and the output is at the top. It is not straightforward because arrows are missing.
Fig 1 is not well explained in the text.
Answer 9 : Added further explanation for Figure 1.
Point 10 : - Fig 2 is one example of the use of CNN + RNN. Is it your figure, your proposal? If not please put a reference of a paper that uses this model.
Answer 10 : Figure 2 is the CNN-RNN structure I know I drew myself.
Point 11 : - "Sign language communicates using hand gestures, not sounds." -> No (see above)
Answer 11 : As in answer 7, I wrote that various information is required.
Point 12 : - This language is used for the deaf. --> this should be said earlier
Answer 12 : It is merged with the previous sentence and placed at the beginning of the section.
Point 13 : - "A language is a combination of words and words to form a sentence. "
Answer 13 : Removed this sentence.
Point 14 : - Figure 3, please put the references of figure 3 (b) and (c)
Answer 14 : Figure 3 (b), (c) has the same source as (a). The source link is "https://github.com/CMU-Perceptual-Computing-Lab/openpose/blob/master/doc/02_output.md", a sub-site of [34]. [34] has been modified to this link.
Point 15 : - with the structure show in Figure 4.--> shown
Answer 15 : Corrected "show" to "shown".
Point 16 : - p6 : Figure 4 should be simplified by putting all the steps on one unique line.
Answer 16 : Figure 4 has been modified.
Point 17 : - process proceeds --> redundant
Answer 17 : Corrected "process proceeds" to "proceeds".
Point 18 : - Figure 5 is not useful. The normalization can be understood without image.
Answer 18 : Figure 5 has been removed. An additional explanation was prepared to replace Figure 5.
Point 19 : - Section 4 should be better written. The beginning is abrupt
Answer 19 : At the beginning of section 4, additional information on how the experiment was conducted was written.
Thanks again for your review.

Round 3
Reviewer 3 Report
Dear authors,
I think you should really improve the quality of the writing.
Several sentences are too simplistic for example the first one "Deep learning is a technology that can dramatically advance the development of cutting-edge technology in various fields ". The whole paper has to be improved.
In addition, for me, the experiments are not convincing enough.